# Development and Validation of a New Gadget Addiction Scale (Screen Dependency Scale) among Pre-School Children in Malaysia

**DOI:** 10.3390/ijerph192416916

**Published:** 2022-12-16

**Authors:** Azwanis Abdul Hadi, Siti Ruziana Roslan, Edre Mohammad Aidid, Nurzulaikha Abdullah, Ramli Musa

**Affiliations:** 1Department of Family Medicine, Kulliyyah of Medicine, International Islamic University Malaysia, Kuantan 25200, Malaysia; 2Department of Community Medicine, Kulliyyah of Medicine, International Islamic University Malaysia, Kuantan 25200, Malaysia; 3Biostatistics and Research Methodology Unit, School of Medical Sciences, Health campus, Universiti Sains Malaysia, Kubang Kerian 16150, Malaysia; 4Department of Psychiatry, Kulliyyah of Medicine, International Islamic University Malaysia, Kuantan 25200, Malaysia

**Keywords:** screen dependency, validation, pre-school children, questionnaire, Malaysia

## Abstract

Background: Excessive screen time in young children is associated with many harmful consequences including screen dependency. Research has shown a worrying prevalence of media-related dependency among adolescents and pre-school children. There are a few available questionnaires among adolescents but none for pre-school children. This study aimed to design and validate a questionnaire to assess screen dependency among pre-school children aged 4 to 6 years old. Methodology: A cross-sectional two-phase study was carried out to develop the scale. In phase 1, a preliminary parent-report measure questionnaire was developed in Bahasa Malaysia. Later, it was sent to four experts for content validity followed by face validity. In Phase 2, a total of 386 parents of pre-school children aged 4 to 6 years old, split into two samples, were involved in the field study for exploratory factor analysis (EFA) and confirmatory factor analysis (CFA). Result: Sample 1 was used to perform EFA to determine the factorial structure of the SDS. All items with a factor loading of >0.4 were included. Sample 2 was used to perform the CFA. RMSEA and CFI analysis showed that the SDS has a good fit and confirms the dimensional structure found via EFA. The final questionnaire consists of 15 items with a 4 factors’ structure and has excellent internal consistency reliability. Conclusions: The Screen Dependency Scale (SDS) is a reliable and valid questionnaire to detect screen dependency among pre-school children aged 4 to 6 years old in Malaysia.

## 1. Introduction

The brains of young children below 6 years old are at the most critical stage of development [1]. Any prolonged environmental exposure such as exposure to digital screens might influence neural growth leading to a variety of consequences [2]. Nowadays, children are using screens extensively at an increasingly younger age with some starting as early as infancy [3,4]. Population surveys have shown that young children use the screen for more than 1 to 2 h per day [5,6].

Excessive screen time in children is associated with many health and psychological consequences such as obesity, language development delays, poorer cognition, behavioural problems, and emotional dysregulations [7,8,9,10,11,12]. More recent studies supported these associations by discovering microstructural changes in the developing brain especially in the tracts involved in language, visual processing, executive functions, and multimodal association [2,13,14]. Because of this, the American Academy of Pediatrics (AAP) recommends young children from the age of 2 to 5 years old should use screens for less than one hour per day; and advises not to expose children less than 18 to 24 months old to any screen media [9].

Screen dependency is another problem associated with the prolonged use of screens, leading to addiction symptoms similar to substance addiction, such as loss of control and negative emotions (such as irritability and anxiety) when the screen is not available [13,15]. To date, the World Health Organization’s (WHO) International Statistical Classification of Diseases and Related Health Problems 11th edition (ICD-11) and the American Psychiatric Association’s (APA) Diagnostic and Statistical Manual of Mental Disorders, Fifth Edition (DSM-5), have placed Internet Gaming Disorder (IGD) in the section of conditions for further research in the DSM-5 [16,17]. Neurologically, IGD has the same brain structure (frontostriatal and frontocingulate) changes similar to other addictions [18,19].

Screen dependency is defined as the problematic use of screen media which covers all types of media transmitted through a screen, such as smartphones, tablets, laptops, televisions, portable video games, and so on [2]. For young children, it is difficult to group them to a specific media since they are generally exposed to all types of screen media [2,20,21]. Because of this, they are continuously exposed to media via the different types of screens that are available to them.

However, research on screen media-related dependencies or addictions has focused primarily on adults, teenagers, and adolescents since most questionnaires were developed for this group. Up to the year 2020, questionnaires for young children were scarce, with a review finding only three questionnaires developed specifically for young children below 10 years old [20,22,23,24]. Out of this, only one was a parent-report measure suitable for children from the age of 4 to 11 years old [22]. Parent reports are quick, easy to use, and cost effective. The concurrent validity of the parent report was high for judging language skills identifying language delay. There were no differences in the rating accuracy of mothers with different educational levels. It was also good at reporting behavioural and conduct problems. As of the date of writing, questionnaires for screen media dependency developed and validated specifically for pre-school children are not available. Hence, not much is known about screen media dependency in this age group, from basic information such as its prevalence in the population and its associated factors. Pre-school children are the group of interest since early exposure and prolonged use at the age of the most rapid brain development could cause complications that are substantial if not detected and subject to early intervention. Hence, this study aimed to develop and validate a screen-media dependency questionnaire for pre-school children.

## 2. Methods

### 2.1. Participants

Our study population were parents (either the father or mother) with children attending registered pre-school centres in Kuantan, Malaysia, and our target population was the preschool children aged 4 to 6 years old attending registered preschool centres in Kuantan, Malaysia. Inclusion criteria were parents of children aged 4 to 6 years old, who were using any type of digital device; Malaysian citizens; and literate in Bahasa Malaysia. Children with other medical disorders such as ADHD, autistic-like behaviour, or congenital disorders were excluded. The data were obtained from 386 children via stratified random sampling based on the list of registered preschools centres.

### 2.2. Procedure

This is a cross sectional validation study, which was divided into 2 phases, and is summarized by the flowchart in Figure 1.

#### 2.2.1. Phase 1

Phase 1 comprised of screening questionnaire formulation, content validity, and face validity. The team that was involved included two family medicine specialists, two psychiatrists, and one public health specialist. The items of the SDS were created based on the IGD criteria in the DSM-5, content from other literature, and clinical experience. A total of 40 items were produced which covered the domains of the child’s preoccupation with screen media; loss of control over child’s screen media use; child’s withdrawal symptoms; child’s craving; social or lifestyle impairment; the dangerous consequences of screen media use; the child’s increasing tolerance; use of deception associated with screen media and physical complaints due to excessive use. The items were developed in statement forms and parent-report style since the target populations were pre-school children. The initial pool of items was reviewed and revised before proceeding to content and face validity. It was also checked by a language expert to ensure the proper and correct use of language.

Four experts were involved in the content validity which were: a child and adolescent psychiatrist; an addiction medicine psychiatrist; a family medicine specialist; and a child psychologist. Expert opinion was given based on the content of the questionnaire, whether it adequately captured screen dependency among pre-schoolers. This included the domain used and the items within each domain. Other than that, whether each item represented the domain or a different domain, whether the item(s) should be included or excluded in the questionnaire, the percentage for each item should be included in the questionnaire based on its significance, and any suggestions for improvement of the questionnaire. Revisions were made based on the feedback given. The preliminary version of the questionnaire after content validity comprised of 30 items distributed over 7 domains.

Face validity was completed next to check whether the item produced was comprehensible for the population and did not contain any medical jargon. This was achieved by sending the questionnaire to five parents/guardians who fulfilled the inclusion criteria and were not involved in the full-scale study later.

#### 2.2.2. Phase 2

Phase 2 included a pre-test, the actual field study (larger population), data collection, and analysis.

A pre-test on 30 participants was carried out to detect any confusion on items, and whether respondents had suggestions for possible improvements of the items. It also gave a rough idea of the response distribution to each item, which was important in determining whether there was enough variation in the response to justify moving forward with a larger-scale test. This also provided feedback on the data collection flow before proceeding with the larger field study.

The larger field study consisted of two samples for factor extraction analysis using exploratory factor analysis and dimensionality via confirmatory factor analysis, respectively [25]. It was recommended that EFA should have a sample size of at least 50 respondents or a respondents-to-items ratio of 5:1 and that CFA should have a sample size of at least 200 respondents [26]. In this study, the first sample consisted of 150 participants and the second sample consisted of 236 participants, which was higher than the recommendations, and which had the advantage, among others, of better item ratio and lower measurement errors [27].

### 2.3. Data Collection Procedure

A list of selected pre-schools was obtained from the State Education Department. The selected pre-schools were approached individually by the researcher. The teacher was briefed regarding the research. Any questions regarding the research were answered before the distribution of the questionnaire. After adequate explanation and consent was given by the pre-school centre, the teacher then shared with the parents the summary of the research and patient information sheet via a WhatsApp^®^ group. Any questions regarding the research were replied by the researcher. Only then, with the help of the teacher, was the link to the questionnaire distributed via the WhatsApp^®^ group. The questionnaire was in Google^®^ forms given the pandemic situation to help minimize visits to pre-school centres. This method also helped parents to interact directly with the researcher as well as ensuring appropriate time for the parent to give their responses. The parent needed to fill in the consent form before they could proceed with the questionnaire. All the responses were automatically sent to the researcher after completion. The data collection was executed from 1 October 2020 to 28 February 2021.

### 2.4. Assessment Materials

This study instrument comprised of 2 sections. Section A was the sociodemographic data of the population. This was to provide additional information regarding the subject of the study regarding the parent, children, environment as well as the media viewing habits of the children in this study.

Section B consisted of the newly developed SDS questionnaire to detect screen dependency disorder. The questionnaire was in Bahasa Malaysia, suitable for the Bahasa Malaysia-speaking population, which is the majority in this country. A Likert scale was used for item scoring. This study used a 4-point Likert scale, ‘Strongly disagree’, ‘Disagree’, ‘Agree’, and ‘Strongly Agree’ with a score of ‘1’, ‘2’, ‘3’, and ‘4’, respectively.

## 3. Statistical Analysis

### 3.1. Exploratory Factor Analysis (EFA)

Exploratory factor analysis (EFA) is an exploratory method. It aims to explore the items, factor common concepts, and generate a theory. Factor extraction for this questionnaire was determined by calculating the factor loading of each item. In the factor analysis procedure, items that possessed similar characteristics were grouped under one component. The suitability for factor analysis was assessed by Bartlett’s test (the assumption that correlation between variables is all 0), the sufficiency of items for each factor, and the Keiser–Meyer–Olkin (KMO) measure should be >0.7 for the item to be analysed for factor loadings. Factor extraction was determined by multiple criteria (eigenvalue > 1, scree plot, parallel analysis, rejection of a factor with fewer than three items, or fixing to a specific number of factors) [28]. Exploratory factor analysis (EFA) was then calculated using IBM SPSS Version 26. Principal axis factoring with Promax rotation was applied for construct validity. Principal-axis factor analysis attempts to simplify common variance amongst a set of different variables. Promax rotation allows factors to be correlated and more useful for large datasets [29]. Factor loading of >0.4 was considered good. This indicated that the item corresponded to the domain. To determine the most parsimonious solution in the measure, two criteria were applied: (a) Kaiser’s rule (eigenvalue > 1.0); and (b) the average of communalities above 0.6 [30].

### 3.2. Confirmatory Factor Analysis (CFA)

Confirmatory factor analysis (CFA) is a confirmatory method based on common factor model. It is a type of Structural Equation Modelling (SEM) analysis that deals with measurement models. It allows for the assessment of measurement model fit. CFA was calculated using IBM SPSS AMOS Version 24. It provided theoretical and empirical justification of the results of preliminary EFA. Comparative fit index (CFI) > 0.9 and root mean square error of approximation (RMSEA) < 0.08 is considered absolute and incremental fit of the model [25]. There is no agreement among researchers on which fitness indexes to use. Hair et al. (1995) recommended the use of at least one fitness index from each category of model fit [31]. If the indexes obtained did not achieve the required level, the factor loading for every item were examined. During the fitness index analysis, items which had a low factor loading were deleted since these items were considered problematic in the model. If the Fitness Index was still not achieved after the low factor loading items have been removed, Modification Indices (MI) were looked at. A high value of MI (>15) indicated that there were redundant items in the model [25].

## 4. Reliability

For this study, the reliability was evaluated by calculating the internal consistency. Internal consistency is commonly estimated using the coefficient alpha, also known as Cronbach’s alpha. Cronbach’s *α* = 0 indicates no internal consistency (i.e., none of the items are correlated with one another), whereas *α* = 1 reflects perfect internal consistency (i.e., all the items are perfectly correlated with one another). A Cronbach’s alpha of at least 0.70 has been suggested to indicate adequate internal consistency [32].

## 5. Results

### 5.1. Sociodemographic Characteristics of the Respondents

The mean age of parents was 35.6 years old while 42.2% of pre-school children were 5 years old. Most of the respondent were mothers (85.2%), had a tertiary level of education (77.2%), and were working (85.5%). There were slightly more female children in this study, with 54.7% female and 45.3% male. Most of them had more than one sibling (80.5%). This study’s respondents were predominantly from urban areas (83.2%), with 86.7% of them having internet facilities at home, and majority of the children used smartphones and televisions (86.5% and 83.9%). Most children (58% on weekdays and 65.8% on the weekends) in this study used more than 1 h of screen time per day. A smaller percentage (4.7%) used more than 4 h of screen time per day, and it tripled (16.8%) over the weekend. The main on-screen activity for pre-school children in this study was watching online videos (84.7%) in excess of playing video games (21%) and using educational apps (49.3%) (please see Table 1).

### 5.2. Exploratory Factor Analysis

The sample was adequate for factor analysis determined by Kaiser–Meyer–Olkin of 0.883 and Bartlett’s Test of Sphericity was significant with a *p*-value of less than 0.05. Initial analysis with Eigenvalue of >1 with a cumulative explained variance of 57% yielded seven-factor structures (Figure 2). However, after Promax rotation, there were three factors with one or two items only that were deleted. The sample was re-run again using a fixed number of four factors. Principal axis factoring using Promax Kaiser normalization for rotation method yielded 21 items with a factor loading of >0.4 grouped under four factors. After considering factor loading, item correlations, and content of the item, nine items were removed sequentially. The final scale was 21 items under 4 domains. The domains included preoccupation with screen media, behavioural issues related to screen media use, effects on daily life activities, and parent’s perception of child’s screen-media tolerance. 

### 5.3. Confirmatory Factor Analysis

The structure model gained from exploratory factor analysis was analysed for confirmatory factor analysis. During the initial confirmatory factor analysis, the fitness index and the measurement error for the model were not acceptable. Thus, any item having a factor loading of less than 0.6 and an R^2^ less than 0.4 were deleted from the measurement model. The final model after deletion of 6 items with low factor loading and R^2^ was 15 items under 4 factors (see Figure 3). A high value of MI (above 15) indicates there were redundant items in the model, thus correlation was made in between the items. The CFA had to fulfil all goodness of fit indices (absolute, incremental, and parsimonious fit) for the latent construct to be considered fit (see Table 2) [26]. Overall, the SDS four-factor model had a good fit and confirmed the dimensional structure found via EFA (see Figure 3).

### 5.4. Reliability Testing

The internal consistency reliability for the final 15 items and 4 domains was good, whereby all domains had a Cronbach alpha of >0.7 while the Cronbach alpha total was 0.899 (Table 3). The sample size in this study is determined by using the subject to item ratio. Thus, the 95% confidence interval was determined for the Cronbach Alpha. The 95% confidence interval ratio was 0.889 to 0.917. Hence, the Cronbach alpha for the whole questionnaire was very good at 0.9 with a 95% confidence interval of 0.889 to 0.917 (Table 4).

### 5.5. Scoring

The possible lowest score for the SDS is 15 and the maximum is 60, the higher scores indicate higher dependence. To determine the performance of the SDS as a screening test, we used the Receiver-Operating Characteristic (ROC) curve to define the best sensitivity and specificity of a cut-off point in the scores (see Figure 4). The ROC analysis is widely used for ascertaining the best cut-off score in many screening and diagnostic tools balancing the fundamental compromises that exist between sensitivity and specificity [33,34]. The cut-off score for the SDS was determined by using the Youden index which maximizes the differences between true-positive and false-positive rates (see Table 5). The Youden index is the point where the sum of sensitivity and specificity was the highest and we calculated ((sensitivity + specificity) − 1) for all scores [33]. To obtain the sensitivity and specificity for all the scores, the total scores of the 15 items in SDS were analysed against the children’s screen time. Based on the ROC, the closer the curve is located to the upper-left hand corner and the larger the area under the curve, the better the test is at discriminating between dependent and non-dependent. The area under the curve (AUC) can have any value between 0 and 1 and it is a good indicator of the goodness of the test. A perfect diagnostic test has an AUC 1.0., whereas a non-discriminating test has an area of 0.5.

The score of 24.50 is the cut-off score which the SDS can detect non-dependence and dependence to the screen, with the sensitivity of 55% and specificity of 80% (see Table 4). The area under the curve (AUC) is an overall summary of diagnostic accuracy [33,34]. The AUC for the SDS is 0.7 (see Figure 4). An AUC of 0.5 suggests random chance, 0.7 to 0.8 is considered acceptable, 0.8 to 0.9 is considered excellent, and more than 0.9 is considered outstanding [34,35].

The mean score for the participants of this study is 25.58, which is above the cut-off score of 24.5. Slightly more participants (50.8%) fall into the non-dependent group, and 49.2% are in the dependant group (see Table 6).

## 6. Discussion

Increased screen time is associated with many negative consequences on children’s physical and mental health. Because of this, the APA recommends that young children from the age of 2 to5 years old only be exposed to good quality media content for less than 1 h per day [9]. The sociodemographic data in this study showed that during weekdays, only a minority of children followed this recommendation, and this number was even less during the weekends. Clearly, the children in this study use the screen more than the current recommendations, which is similar to other local and international population studies [5,6].

In recent times, the advancement of the internet has allowed for media content to be streamed at any time, making this a preferred choice of media delivery as compared to traditional media (television broadcast, cable, and satellite). Hence the popularity of digital-ready devices such as mobile phones which can access digital media at any time and place. However, modern television is now equipped with internet capabilities that can be used to watch online media. Therefore, despite the increasing utilization of smartphones among young children, the television continues to also be utilized by young children [3]. This is consistent with the finding that most of the children in this study used the smartphone and the television as opposed to other screen devices. This is also in line with the finding that the main on-screen activity for pre-school children in this study is watching online videos, playing video games, and using educational apps. The combination of the availability of both continuous media content and digital-ready devices, which are available both on the go and at home, contributes to the prolonged and excessive use of the screen which could lead to screen dependency.

The main purpose of this study was to develop a sound and reliable psychometric screening questionnaire to assess screen dependency among pre-school children. As elaborated above, the SDS underwent a robust psychometric analysis and demonstrated adequate results on the construct validity and excellent internal consistency, proving that the SDS is a valid and reliable tool to be used.

The SDS was developed based on the DSM-5’s IGD criteria and other literature on addiction [16,36,37]. The final 15 items under 4 domains covered important areas in the DSM-5’s IGD. The four domains are “Preoccupation with screen media”, “Behavioural issues related to screen media use”, “Effect on daily life activities”, and “Parent’s perception on child’s screen media tolerance”. The “Preoccupation with screen media” domain reflects the child’s obsession with the screen to the extent that he/she asks for the screen at every possible moment. The “Behavioural issues related to screen media use” domain reflects the withdrawal criteria in DSM-5’s IGD. In behavioural addiction, withdrawal manifests primarily as symptoms, with lesser physical signs as seen in pharmacological withdrawal [38]. Symptoms such as irritability or anxiety from a young child, either from the use of the screen or from the effect of stopping the screen itself, can be obtained from the parent’s observations of the child’s behaviour. The questions in this domain also capture the negative behaviour that occurs from the difficulty in controlling the child’s screen use. The “effect on daily life” domain covers one of the most important areas in screen dependency: excessive use to the extent of affecting the normal functioning of daily living [2,21]. The DSM-5 notes the severity of dependency depends on the degree of daily activities that are affected [16]. The final domain covers tolerance which is the need to use the screen longer or more frequently to obtain the same satisfying effect [38]. The sentence structure in this domain was constructed based on the parents’ perception since tolerance is difficult to determine from a third person. However, in a young child who is still dependent on their parents, the parents’ observation of their child’s activities is legitimate, as seen in other studies on the accuracy of parental observations [39,40]. The SDS covers important domains that capture screen dependency in children, within the scope of 15 items, which can be answered within 5 min, making it a suitable tool to be used for busy parents and in combination with other questionnaires in future epidemiological studies.

The SDS can be scored from the lowest to the highest possible scores of 15 to 60, with the higher score meaning higher dependency to the screen. The mean score for the participants of this study is 25.58, which is above the cut-off score of 24.5. Slightly more participants (50.8%) fall into the non-dependent group, and 49.2% are in the dependant group.

This study has its limitations. The SDS was developed in Bahasa Malaysia and is a good questionnaire to be used among Malaysians that are well-versed in this language. Therefore, it is recommended that this questionnaire be validated in other languages and cultures. This study also suffers from recall and social desirability bias, which is a common limitation of self- or parent-report measures. Apart from that, this study did not undergo further validation to assess its relationship with other variables/measures, such as against a gold standard (criterion validity) or other similar measures (convergent validity). This is due to the fact that there is no gold standard for screen dependency diagnoses at this moment. However, future research can include further validation against clinical diagnosis once a standardized diagnostic criterion is established.

There is heated debate among researchers on whether categories are acceptable in psychological research since some psychological disorders such as personality disorders or pathological worry degree of severity exist as a unidimensional spectrum and categorizing may mask its underlying reality, while other psychological disorders such as addiction, depression, or eating disorders have a clear latent construct to differentiate normal and abnormal [41,42]. Statistically, it is also recommended that scores of psychological illnesses be best interpreted in continuous data rather than in categories to minimize errors [43,44]. However, it is also argued that it is practical to have cut-off scores to facilitate the clinical assessment and for comparison in future epidemiological studies [41]. The cut-off in this study, again, is suitable for this population and it is recommended to compare with cut-off scores elicited in clinical samples or with other similar questionnaires for further validation.

## 7. Conclusions

The Screen Dependency Scale (SDS) underwent a robust psychometric methodology involving questionnaire development, content validation, face validation, pre-testing, and internal structure evaluation with EFA and CFA. It was found to be a reliable and valid tool to be used in future epidemiological or clinical studies in Malaysia and the results would give further information on screen dependency among children that would initiate further development in this area, such as the formulation of a screening algorithm and intervention. This questionnaire, once translated and validated accordingly, can also be used in other populations.

## Figures and Tables

**Figure 1 ijerph-19-16916-f001:**
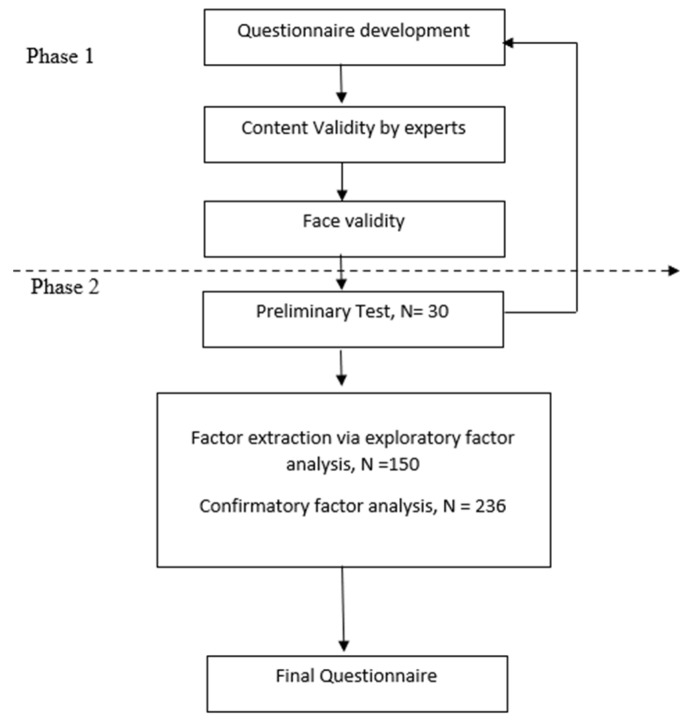
Flowchart of the development and validation process.

**Figure 2 ijerph-19-16916-f002:**
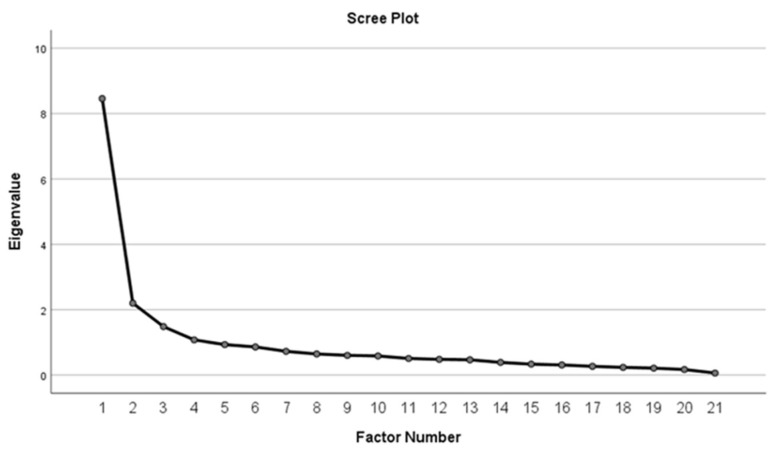
Scree plot.

**Figure 3 ijerph-19-16916-f003:**
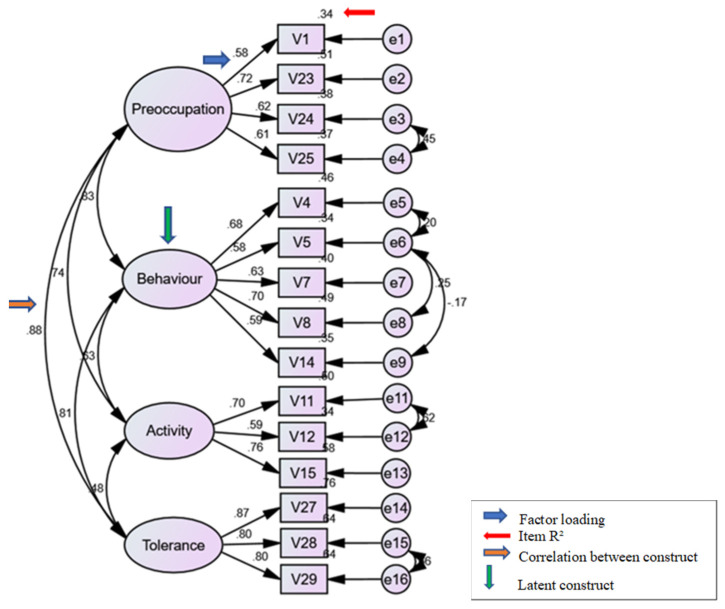
Final questionnaire factor structure model by AMOS SPSS.

**Figure 4 ijerph-19-16916-f004:**
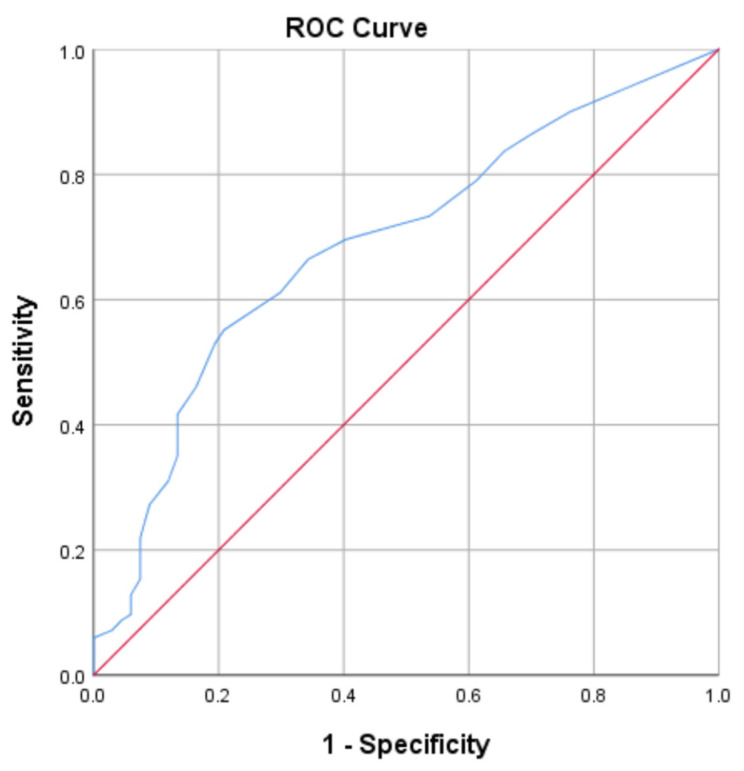
ROC curve for the Screen Dependency Scale. AUC, area under the curve; ROC, receiver operating characteristic. AUC = 0.7.

**Table 1 ijerph-19-16916-t001:** Sociodemographic of the parent, the children involved, as well as their media viewing habits.

Variable	*n* (%) or Mean (SD)
**1.** **Parent/guardian information**	
** *Relationship with child:* **	
Mother	329 (85.2)
Father	54 (14)
Guardian	3 (9)
** *Ethnic:* **	
Malay	362 (93.8)
Chinese	16 (4.1)
Indian	1 (3)
Others	8 (2.3)
** *Religion:* **	
Islam	368 (95.3)
Buddhist	16 (4.1)
Hindu	0 (0)
Others	2 (5)
** *Age* **	35.69
** *Education level:* **	
No formal education	1 (3)
Primary school	0 (0)
Secondary school	87 (22.5)
University/College	298 (77.2)
** *Working status:* **	
Yes	330 (85.5)
No	56 (14.5)
**2.** **Child information**	
** *Age (Years)* **	
4	97 (25.1)
5	163 (42.2)
6	126 (32.6)
**Sex:**	
Male	175 (45.3)
Female	211 (54.7)
** *Number of siblings:* **	
Only child	58 (19.5)
Others	328 (80.5)
**3.** **House environment**	
** *Housing area:* **	
Urban	321 (83.2)
Rural	65 (16.8)
** *Internet at home:* **	
Yes	338 (87.6)
No	48 (12.4)
**4.** **Child’s media viewing habits**	
** *Screen device use:* **	
Smartphone	300 (86.5)
TV	291 (83.9)
Tablet	81 (23.3)
Laptop	79 (22.8)
Video games	17 (4.9)
** *Screen activity:* **	
Watching online videos	294 (84.7)
Playing video games	73 (21)
Using educational apps	171 (49.3)
Surfing the internet	49 (14.1)
** *Screen time per day (Weekday)* **	
<1 h	144 (37.3)
1–4 h	224 (58.0)
>4 h	18 (4.7)
** *Screen time per day (Weekend)* **	
<1 h	67 (17.4)
1–4 h	254 (65.8)
>4 h	65 (16.8)

**Table 2 ijerph-19-16916-t002:** Summary of the Screen Dependency Scale’s goodness of fit.

Model	Absolute Fit	Incremental Fit	Parsimonious Fit	Level of Acceptance
RMSEA	0.085	-	-	<0.08 (fair fit), >0.10 (poor fit) [27]
GFI	0.90	-	-	>0.9 (satisfactory fit) [27]
CFI	-	0.93	-	>0.9 (acceptable fit) [26]
Chisq/df	-	-	2.70	<3.0 (satisfactory fit) [26]

**Table 3 ijerph-19-16916-t003:** Summary of Screen Dependency Scale’s items (translated), factor load, and reliability.

Factor Name	Item Number	Item	Factor Load	Cronbach’s Alpha (Sub-Scale)	Cronbach’s Alpha (Scale)
Preoccupation with screen media	V1	My child uses the screen longer than the duration that I allowed.	0.53	0.76	0.90
	V23	My child often sleeps late because he/she was using the screen.	0.45		
	V24	My child will ask for the screen immediately after waking up from sleep.	0.72		
	V25	My child will ask for the screen as soon as he/she returns home from school.	0.85		
Behavioural issues related to screen media use	V4	My child finds it hard to stop using the screen.	0.48	0.78	
	V5	My child will only be calm after he/she is allowed to use the screen.	0.69		
	V7	My child will protest (becomes angry, cries or sulk) if he/she was not allowed to use the screen.	0.62		
	V8	My child will take a long time to calm down when he/she was denied using the screen.	0.68		
	V14	My child will easily cry/sulk/becomes angry if they use the screen for too long.	0.73		
Effect on daily life activities	V11	My child has not attended school because he/she slept late because of using the screen.	0.76	0.75	
	V12	My child has not attended school because he/she was occupied with the screen.	0.84		
	V15	My child’s sleeping pattern has changed because of screen use.	0.46		
Parent’s perception on child’s screen media tolerance	V27	I feel that my child is using the screen excessively.	0.51	0.90	
	V28	I feel that my child is using the screen more frequently than before.	0.86		
	V29	I feel that my child is using the screen longer than before.	0.86		

**Table 4 ijerph-19-16916-t004:** Confidence interval for Cronbach Alpha.

Cronbach Alpha	95% Confidence Interval	
	Lower Bound	Upper Bound
0.904	0.889	0.917

**Table 5 ijerph-19-16916-t005:** Some of the coordinates (scores) of the curve; the sensitivity and specificity of the scores; and the Youden index.

SDS Scores	Sensitivity	1 − Specificity	Specificity	Sensitivity + Specificity − 1
14.00	1.0	1.000	0.000	0.00
15.50	0.94	0.87	0.13	0.08
16.50	0.90	0.76	0.24	0.14
17.50	0.87	0.70	0.30	0.16
18.50	0.84	0.66	0.34	0.18
19.50	0.79	0.61	0.39	0.18
20.50	0.73	0.54	0.46	0.20
21.50	0.70	0.40	0.60	0.30
22.50	0.67	0.34	0.66	0.32
23.50	0.61	0.30	0.70	0.31
**24.50**	**0.55**	**0.21**	**0.79**	**0.34a**
25.50	0.53	0.20	0.81	0.34
26.50	0.46	0.16	0.84	0.30
27.50	0.42	0.13	0.87	0.28
28.50	0.35	0.13	0.87	0.22

Youden index: the highest score of the sum of sensitivity and specificity.

**Table 6 ijerph-19-16916-t006:** The mean score and prevalence of screen dependency.

	Mean (SD)	n (%)
Mean score	25.58 (7.663)	
Non-dependent		196 (50.8)
Dependent		190 (49.2)

## Data Availability

The data presented in this study are available on request from the corresponding author.

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
