# Peer review of "Development and Validation of a New Gadget Addiction Scale (Screen Dependency Scale) among Pre-School Children in Malaysia"

_ijerph, 2022, doi:10.3390/ijerph192416916_

Round 1

Reviewer 1 Report

This paper is worthy to be published and writing is very concise and clear. I have a few comments below.

-      Method: Please state the date of collection (i.e, May 2022-September 2022)

-      p.3 Family Medicine Specialists, Psychiatrists, Public Health Specialist, Child and Adolescent psychiatrist, Addiction medicine etc.  à please the first letters which was capitalized was needed to change into small letters. (i.e., F, M, S à state small letter, family medicine specialist).

-      p.3 line 116 a few parents/guardians à Please state the exact numbers of parents/guardians.

-      p.7 line 217 & 218, Please change R2à R2

Reviewer 2 Report

Thank you for the opportunity to review this manuscript titled “Development and validation of a new gadget addiction scale (Screen Dependency Scale) among pre-school children in Malaysia”. The authors designed and validated a questionnaire to assess screen dependency among pre-school children. The topic of the study was really interesting and important and has practical implications. There are some concerns should be addressed before publication.

Page 2 lines 52-58: I think these two sentences have little relevance to the theme. In addition, the authors need to describe the current situation and characteristics of screen dependence of children aged 4-6 years, as well as the similarities and differences between them and older children, in the Introduction section.

Page 2 lines 77-84: The authors mentioned that “Participants were a parent (either the father or mother) of preschool children aged 4 to 6 years old attending registered preschool centres in Kuantan, Malaysia.” This sentence is a bit vague, I think participants were preschool children father or mother more simple and clearer. Besides, are these parents willing to participate in the study? The authors need to describe in detail the inclusion criteria and exclusion criteria of the subjects.

Page 9 line 233: The authors said they found a four-factor structure. I suggest the authors explain the name of each dimension and summarize the contents of each dimension. I have a little doubt that the authors did not provide criterion related validity, which should be provided when developing a scale. Furthermore, the authors need to do a follow-up study to provide the test-retest reliability of the scale.

Page 10 lines 258-261: The authors found that slightly more participants (50.8%) fall into the non-dependent group. So, I suggest the authors do some analyze, such as the scores of the scale and subscales, as well as demographic characteristics of the subjects. I see, part of the results presented in the Discussion. This kind of writing is a little unreasonable.

Reviewer 3 Report

Thank you for giving me this opportunity to review this manuscript, “Development and validation of a new gadget addiction scale (Screen Dependency Scale) among pre-school children in Malaysia”. This study aimed to design and validate a questionnaire to assess screen dependency among pre-school children aged 4 to 6 years old. While the manuscript was generally well-written, I have some concerns about key study information that appears to be missing from the manuscript; and, some clarification from the authors regarding the data set and analyses are also recommended. My comments are outlined below and I sincerely hope that the authors find them helpful in any future revisions of their work.

1. Abstract

You stated that “Sample 1 was 24 used to perform EFA to determine the factorial structure of the SDS; Sample 2 was used to perform the CFA. I would suggest to place these sentences in the part of methodology rather than illustration in the reults parts. Also, the abbreviation for The Screen Dependency Scale should appear first, not at the conclusion.

2. Introduction

The introduction is well-written. I would only suggest to elaborate more regarding the reason that why it is important of parent-report on measuring children’s screen dependency. As you stated in the manuscript, “out of this, only one is a parent-report measure suitable for children from the age of 69 4 to 11 years old”, why would this method matter? It is suggested to give more explanation on this to the readers.

3. Method-participants

The description of your subjects is not very clear. It is easy to confuse readers whether the questionnaire is administered to parents or children.

4. Method- data analysis

Regarding to the evaluation of CFA, SRMR is an important index. However, you did not include this index, why? Moreover, the convergent and discriminant validity are required to examine on CFA. You could consider to include these in your study. Besides, the analysis of Receiver-Operating Characteristic (ROC) should be also explained in the data analysis section.

5. Result-EFA

In your results of EFA, I would suggest add the figure of scree plot which is useful for the readers to understand how many factors should be extracted.

6. Results-CFA

You stated that “a high value of MI (above 15) indicates there were redundant items in the model, thus correlation was made in between the items”. I can not understand why higher MI reflects the redundant items. Also, I also can not understand why the correlation of the measurement error are needed to address this situation.

Reviewer 4 Report

The current study developed and validated a screen media dependency questionnaire for pre-school children. This provides a potentially useful tool in future studies investigating the psychopathology of screen use dependence. The manuscript is clearly written.

Several concerns/suggestions are listed below:

L67-L70: The authors states “Questionnaires for young children are scarce… only one is a … suitable for children from the age of 4 to 11 years old”, which is logically important for this study’s novelty. However, even a quick search for literature shows there are studies with similar attempt (e.g., Yalçin et. al., BMC Pediatrics 2021, https://doi.org/10.1186/s12887-021-02939-y). The authors are suggested to review more carefully about literatures on this topic and revise instruction as needed.

L128-L129: Acronym of “EFA” and “CFA” in the main text should be defined first.

L202: in Table-1, variable “gender” is used. According to The AMA Manual of Style, Sex refers to the biological characteristics of males and females, and is defined as the classification of living things as male or female and is a “biological component, defined via the genetic complement of chromosomes, including cellular and molecular differences.

Whereas gender includes more than sex and serves as a cultural indicator of a person’s personal and social identity. 

Please review and check for your usage of gender and/or sex.

References:

AMA Manual of Style: A Guide for Authors and Editors (11th ed.)

Clayton JA, Tannenbaum C.  Reporting sex, gender, or both in clinical research?  JAMA. 2016;316(18):1863-1864.

L217: How was the factor loading threshold of 0.6 determined? What do authors mean R2<0.4 mean here?

Figure 2 needs a figure legend.

L252-L257: in testing sensitivity/specificity of multiple thresholds/cut-off, it was not clear by reading these paragraphs how to designate “true dependent” or “true non-dependent”, please clarify.

Round 2

Reviewer 2 Report

The authors replied to most of my comments and pointed out that some content will be completed in future research. Therefore, I think this article is acceptable for publication.

Reviewer 3 Report

Dear authors

Thanks for your effort on the revision. Congratulation to your work.